

# The characteristics and roles of antimicrobial peptides as potential treatment for antibiotic-resistant pathogens: a review

Nurul Hana Zainal Baharin[1,2]   Nur Fadhilah Khairil Mokhtar[2]   Mohd Nasir Mohd Desa[1,2]   Banulata Gopalsamy[1]   Nor Nadiha Mohd Zaki[2]   Mohd Hafis Yuswan[2]   AbdulRahman Muthanna[1]   Nurul Diana Dzaraly[1]   Sahar Abbasiliasi[2]   Amalia Mohd Hashim[2,3]   Muhamad Shirwan Abdullah Sani[4]   Shuhaimi Mustafa[3]

[1] Department of Biomedical Science, Faculty of Medicine and Health Sciences, Universiti Putra Malaysia, Serdang, Selangor, Malaysia

[2] Laboratory of Halal Science Research, Halal Products Research Institute, Universiti Putra Malaysia, Serdang, Selangor, Malaysia

[3] Department of Microbiology, Faculty of Biotechnology and Biomolecular Science, Universiti Putra Malaysia, Serdang, Selangor, Malaysia

[4] International Institute for Halal Research and Training (INHART), International Islamic University Malaysia, Gombak, Selangor, Malaysia

Corresponding author
Mohd Nasir
Mohd Desa, mnasir@upm.edu.my

## ABSTRACT

The emergence of antibiotic-resistant bacteria has become a significant and ever-increasing threat to global public health, increasing both morbidity and mortality rates, and the financial burden on health services. Infection by drug-resistant bacteria is anticipated to contribute to the demise of almost 10 million people by the year 2050 unless a competent and effective response is devised to engage with this issue. The emergence and spread of resistance are commonly caused by the excessive or inappropriate use of antibiotics and substandard pharmaceuticals. It arises when pathogens adapt to different conditions and develop self-defence mechanisms. Currently, novel antimicrobial peptides (AMPs) have been reported to be the sole cure for some clinical cases of infectious diseases such as sepsis and skin infections, although these agents may, on occasion, require administration together with an adjunctive low-dose antibiotic. Although AMPs are a promising alternative form of anti-microbial therapy and easily applied in the medical sector, they still have limitations that should not be taken lightly. Hence, this review explores the characteristics, advantages and disadvantages of AMPs for their potential in treating antibiotic-resistant pathogens.

# INTRODUCTION

Antibiotics are anti-bacterial medications that inhibit or kill the growth of bacteria. It is usually competent in the treatment of pathogenic bacteria. However, antibiotics constantly lose their anti-bacterial strength as drug-resistant bacteria emerge with the misuse of the

antibiotics (*Gould & Bal, 2013*; *Sengupta, Chattopadhyay & Grossart, 2013*; *Wright, 2014*). The antibiotic-resistant bacteria are evolving worldwide, threatening the effectiveness of antibiotics, which have previously recovered millions of lives from infectious diseases (*Golkar, Bagasra & Pace, 2014*; *Gould & Bal, 2013*; *Sengupta, Chattopadhyay & Grossart, 2013*; *Wright, 2014*). This scenario is becoming an important public health problem as it will lead to a prolonged hospital stay, increasing the cost of health care and risk of deaths (*Golkar, Bagasra & Pace, 2014*; *Ventola, 2015*).

The finding for alternative antimicrobial agents with new mechanisms of action is of urgent need. Although some antibiotics are still effective in killing bacteria, long-term concerns about the good and bad effects of their usage remain to be taken seriously. Worryingly, the number of deaths caused by bacterial infections has risen dramatically, making it one of the leading factors of life-threatening diseases (*Morehead & Scarbrough, 2018*; *World Health Organization, 2020*). This problem arises due to the emergence of resistant infectious agents which is a major problem in the treatment of microbial infections (*Ventola, 2015*). The attempts to discover other substances to replace the function of available antibiotics are still on going and being explored up to this day.

Antimicrobial peptides (AMPs) are one of the alternative components that may inhibit bacterial growth, possibly replacing the function of antibiotics in the future (*Pfalzgraff, Brandenburg & Weindl, 2018*). These peptides can bind and interact with the negatively charged bacterial cell membranes, resulting in the disruption of the bacterial cell membrane. They cause damage to the cellular membrane, affecting the transportation of the large molecules such as proteins and ruining the morphology of the cells leading to cell death (*Lei et al., 2019*). Besides, some AMPs such as non-lytic AMPs including buforin II, indolicin and drosocin can also translocate across bacterial membrane to act on intracellular targets, including ribosomes (*Cardoso et al., 2019*; *Le, Fang & Sekaran, 2017*). These provide a good rationale for AMPs as a potential alternative for treatment of antibiotic-resistant pathogens. This paper provides an overview on the dilemma and impact of antibiotic resistance, followed by naturally occurring AMPs and synthetic AMPs analogues as a potent antimicrobial agent, in view of finding solutions and improving the quality of health worldwide. The advantages and disadvantages of AMPs are also discussed in relation to those of antibiotics. This paper is intended for all scientists and academicians in related fields to recognize the potential use of AMPs as a therapeutic agent and to serve as a reference for future related studies.

## SURVEY METHODOLOGY

We conducted a literature search covering publications in 2011 till 2021 in relevant topics. The keywords used in the search included ''antimicrobial peptides'', ''antibiotic-resistance'', ''mechanisms of action'', ''AMPs'', ''bacteriocins'', ''bacterial resistance'', ''mode of action'', ''multi-drug resistance'', ''naturally occurring AMPs'' and ''synthetic AMPs'' through Google Scholar, Web of Science and PubMed Central platform. To ensure a comprehensive and unbiased coverage of the literature, all papers were assessed for information related to the crisis of antibiotic resistance; the history, sources and structure

of AMPs; the bacterial resistance mechanisms and mechanisms of action of AMPs; and the benefits or limitations from the use of antibiotics and AMPs. The assessment of the data was performed by multiple individuals for articles in the different sections, followed by compilation and revision by the first and the corresponding author.

## THE DILEMMA AND IMPACT OF ANTIBIOTIC RESISTANCE

Although the world is rapidly moving towards an era of globalization, especially in the field of medical technology, the problem of antibiotic resistance is something that cannot be denied when the number of deaths caused by bacterial infections has risen dramatically, making it as one of the leading factors of life-threatening diseases (*Morehead & Scarbrough, 2018*). In early 1945, during the era of the discovery of penicillin, Sir Alexander Fleming had proclaimed and warned that antibiotics would one day be a highly demanded drugs and an era of abuse would emerge in the future (*Spellberg & Gilbert, 2014*). Now, many cases of the microbial resistance against antibiotics are being reported year by year. For example, Gentamicin-Resistance *Enterecoccus* (GRE) was first reported to be resistant to vancomycin, the drug that is in use to treat Methicillin-resistant *Staphylococcus Aureus* (MRSA) and Methicillin-resistant coagulase-negative Staphylococci (MR-CoNS) (*Sengupta, Chattopadhyay & Grossart, 2013*).

The pharmaceutical industries have introduced many new antibiotics in the late 1960s to 2000s which include imipenem, ceftazidime, levofloxacin, linezolid, daptomycin and ceftaroline. However, over time, more and more bacterial resistance appears, and the number of new drug discovery steeply decreases. As a result, bacterial infection becomes a great threat to human health (*Ventola, 2015*). Worryingly, the genetic traits for antibiotic resistance can be transferred to other bacteria through horizontal gene transfer (HGT). Some resistance can also be caused by mutations at the genetic level of a bacterial cell leading to expression of altered target sites which is no longer recognized by the antibiotics. This has led to the difficulties in controlling bacterial infections, as many antibiotics will not be able to exhibit similar effect over time and in different individuals (*Read & Woods, 2014*).

To avoid the adverse effects, the use of antibiotics should follow the guidelines in managing infections. Excessive use of antibiotics and misuse can lead to many complications such as diarrhea, indigestion, nausea, yeast infections or digestive problems. Incomplete use of antibiotics will not help killing the germs effectively but will increase the selection of resistance to strive and occupy the niche left by the susceptible strains. Antibiotics can also be toxic if not taken correctly and could turn out to be hazardous if taken more than the recommended doses (*Ventola, 2015*).

Previously, a high percentage of antibiotic resistance was found in farm animals and reached consumers through meat products (*Bartlett, Gilbert & Spellberg, 2013*), resulting in the spreading of bacterial resistance to human and adversely affecting human health (*Centers for Disease Control and Prevention, 2013*). The use of antibiotics in agriculture also affects the microbial balance of the environment (*Bartlett, Gilbert & Spellberg, 2013*). Certain amount of antibiotics ingested by livestock are excreted in their stools or urine.

This causes widespread dissemination to the environment when their stools are used as fertilizers, which can be absorbed into groundwater and even soil (*Bartlett, Gilbert & Spellberg, 2013*). Moreover, antibiotic namely tetracycline is used as pesticides on plants. This practice causes long-term adverse ecological consequences due to the increase of resistant bacteria contaminating the environment (*Golkar, Bagasra & Pace, 2014*).

The crisis of antibiotic resistance which is becoming increasingly pervasive today is having a deleterious effect on the disease management. In general, infections caused by resistant bacterial strains are more severe than those by susceptible strains (*Cosgrove & Yehuda, 2003*). For instance, a significantly higher fatality rate has been reported for MRSA in comparison with that for methicillin-susceptible *S. aureus* infection (*Cosgrove & Yehuda, 2003*; *Engemann et al., 2003*). The adverse effects of antibiotic resistance can be evaluated in accordance with several factors, including an increase in patient mortality, greater resource utilisation, an escalation in the cost of care and reduction in hospital activity (*Friedman, Temkin & Carmeli, 2016*).

Mortality is the most severe consequence of antibiotic resistance. A report by Centers for Disease Control and Prevention (CDC) in 2019, suggests that more than 2.8 million antibiotic resistance associated infections occur in the United States every year, with over 35 000 fatalities. The annual increase in the number of cases has rendered the use of additional isolation rooms and consumable items necessary, with comparable increases being required in nursing care, support services and associated medical tests (diagnostic test and imaging), and these have placed a greater resource utilisation and the overall cost of care (*Friedman, Temkin & Carmeli, 2016*). In the wake of this, a relatively substantial amount of hospital spending, in excess of $4.6 billion annual basis, was reported for the treatment of patients infected by antibiotic resistant bacteria (*Centers for Disease Control and Prevention, 2019*). In addition, this crisis has been observed to rein in everyday hospital activities, with elective operations being cancelled against a background of outbreaks of antibacterial-resistant illnesses (*Macraea et al., 2001*).

## THE ARCHIVAL AND THE DIVERSITY OF AMPS

AMPs have been in the focus as the alternative in addressing the problem of antibiotic resistance. AMPs have been available naturally and synthetically; the history of naturally occurring AMPs and the evolution of synthetics AMPs are discussed in this topic and the analogies between these two forms of AMPs are compiled in Table 1.

### The history of naturally occurring AMPs

Naturally occurring AMPs that act as host defences are found in nearly all forms of life, and most of them have been reported to be isolated from eukaryotes, such as animals, plants and fungi (*Kumar, Kizhakkedathu & Straus, 2018*). AMPs were also found in prokaryotic cells when antimicrobial substances known as gramicidins were isolated from *Bacillus brevis* (*Nakatsuji & Gallo, 2012*). Historically, bacteria have been among the earliest sources of AMPs, and the percentage of AMPs isolated from bacteria have the potential to increase in the future (Fig. 1). In 1939, Dubos extracted AMPs from *Bacillus* strain in soil to protect mice from pneumococcal infections (*Dubos, 1939*). In a previous study, gramicidin

**Table 1 The analogies between Naturally Occurring AMPs and Synthetic AMPs.**

| | Naturally occurring AMPs | Synthetic AMPs | References |
|---|---|---|---|
| Sources/Origin | -Found in many tissues of many different species<br>-Found in nearly all forms of life, and mostly reported to be isolated from eukaryotes, such as animals, plants and fungi<br>-Found in prokaryotic cells | -Non-natural sources<br>-Often created by mimicking natural sequences | (*Jiang et al., 2021*; *Kumar, Kizhakke-dathu & Straus, 2018*; *Nakatsuji & Gallo, 2012*) |
| Content | -Comprised of L-amino acids recognizable by proteases | -The rational design of sequences comprising analogous D-amino acids substituted for L-amino acids | (*Da Cunha et al., 2017*; *Zhao et al., 2016*) |
| Discovery methods | -Using classic purification and *in vitro* and *in vivo* techniques | -Combination of trial-and-error experimentation, screening, or computer-aided design (increasing the peptide post-translational stability without altering biological function). | (*Da Cunha et al., 2017*; *Jiang et al., 2021*) |
| Characteristics | -Frequently susceptible to protease degradation<br>-Low bioavailability (i.e., presence of bioactive molecules at usually low levels).<br>-Low resistance to proteolytic degradation resulting in shorter half-lives | -High bioavailability<br>-Longer half-lives *in vivo*, while maintaining a similar activity and selectivity.<br>-Designed to improve their potential without side effects<br>-Incorporation of multiple functions in the same peptide sequence | (*Azmi, Skwarczynski & Toth, 2016*; *Da Cunha et al., 2017*; *Jiang et al., 2021*; *Lei et al., 2019*; *Lu et al., 2020*; *Mahla-puu et al., 2016*; *Wimley, 2019*) |
| Examples | -Protegerin<br>-Indolicin<br>-Magainin 2<br>-*Moringa oleifera* chitin-binding protein (*Mo*-CBP) | -Iseganan (protegerin as template)<br>-Omiganan (developed from indolicin)<br>-Pexiganan (developed from magainin 2)<br>-*Mo*-CBP$_3$-PepIII (developed from Mo-CBP) | (*Ge et al., 1999*; *Gottler & Ramamoor-thy, 2009*; *Oliveira et al., 2019*; *Sader et al., 2004*; *Trotti et al., 2004*) |

showed antibacterial activity against various Gram-positive bacteria (*Dubos & Hotchkiss, 1941*). Gramicidin is effective in the treatment of infected wounds on guinea pig skin and in the treatment of topical wounds and ulcers (*Van Epps, 2006*; *Gause & Brazhnikova, 1944*), thus demonstrating their potential as the first commercially used AMPs in the health industry. After that, in 1941, other AMPs isolated from bacteria, called tyrocidines, were found to be effective against Gram-negative and Gram-positive bacteria (*Dubos & Hotchkiss, 1941*).

In bacteria, AMPs help particular organisms by killing other bacterial species that compete for the same nutrients and ecological niche. Known as bacteriocins, bacterial AMPs can be classified into two classes: lantibiotics and non-lantibiotics. Lantibiotics are AMPs comprising the non-natural amino acid lanthionine. In 1947, a type of lantibiotic AMPs isolated from *Lactococcus lactis*, known as nisin, was found to be active against a number of Gram-positive bacteria and historically used as a preservative for many years without any noticeable growth of resistance (*Mattick & Hirsch, 1947*). Meanwhile, non-lantibiotics are AMPs composed of thermostable peptides which do not contain lanthionine and do not undergo post-translational modifications (*Heng & Tagg, 2006*).

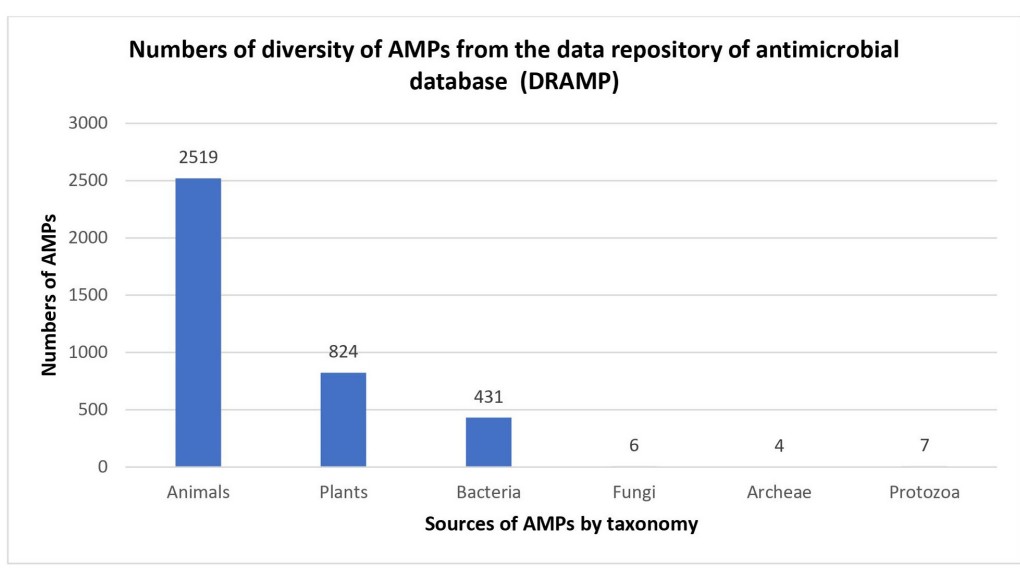

**Figure 1** **Numbers of diversity of AMPs from the data repository of antimicrobial peptides (DRAMP).** Data obtained from http://dramp.cpu-bioinfor.org/browse/.

Garvicin Q (GarQ) is a type of non-lantibiotic AMPs with a relatively broad antimicrobial spectrum towards *Listeria* and *Lactococcus* spp (*Tymoszewska et al., 2017*).

In plants, AMPs play an important role in their protection against the infection of bacteria or fungi. In 1942, another AMPs, called purothionin, isolated from *Triticum aestivum* plants (*Balls, 1942*), was detected to be effective against other bacteria (*Ohtani et al., 1977*). The thionin family is among the best-studied groups of AMPs isolated from plants, apart from plant defensins and cyclotides.

The highest diversity of AMPs was found in animals. For example, in 1956, AMPs named defensins were discovered from rabbit leukocyte isolation (*Hirsch, 1956*). Later, in 1960, an AMP called lactoferrin was successfully isolated from cow's milk (*Groves, 1960*), followed by the synthesis of the AMPs known as bominins from epithelial cells in 1962 (*Kiss & Michl, 1962*).

Several discoveries of AMPs from leukocytes have also been documented around 1970s and 1980s. Among these are rabbit-human α-defensins and purothionin (*Selsted, Szklarek & Lehrer, 1984*; *Selsted et al., 1993*). In 1980, *Hultmark et al. (1980)* used silk butterflies as a model system to successfully demonstrate that P9A and P9B could be induced in the hemolymph by co-vaccination with *Enterobacter cloacae*. Shortly thereafter, these peptides were renamed as cecropin until they became known as the major α-helical AMPs (*Hultmark et al., 1980*). In 1987, *Zasloff (1987)* isolated and characterized cationic AMPs from the African toad frog, *Xenopus laevis*, and named them magainin peptides. A few years later, β-defensin and θ-defensin were characterized after isolation from bovine granulocytes and from leukocytes of the rhesus monkey, respectively (*Diamond et al., 1991*; *Tran et al., 2002*).

In the early 1990s, there were several views that lysozyme was one of the first AMPs to exhibit antimicrobial activity involving non-enzymatic mechanisms. Based on these views, AMPs are seen to have a role in the immunity of human that lacks an adaptive immune system (*Diamond et al., 2009*). In the mid-1990s, several other peptides were also discovered, such as the first anionic AMPs found in *X. laevis*, while other peptides in the rumen of sheep and cattle were characterized (*Brogden, Ackermann & Huttner, 1997*). In addition, AMPs have also been found in fruit flies, called *Drosophila melanogaster*; by losing the genes encoding for AMPs in fruit flies will make them susceptible to fungal infections. This shows the importance of AMPs in protecting flies from microbial invasion (*Lemaitre et al., 1996*).

There are a lot of studies on AMPs that had been conducted to determine their ability to kill bacteria and fight infections. AMPs exist in almost all multicellular organisms and play roles in the mammalian immune system (*Lemaitre et al., 1996*). They have been widely identified in many areas of the human body that are usually exposed to germ-like infections. AMPs are important in innate modulation, as they can be produced naturally by various types of blood cells, including neutrophils, eosinophils and platelets, in the event of inflammation or injury, supporting that AMPs are among the agents responsible for fighting infections caused by germs (*Diamond et al., 2009*).

## The evolution of synthetic AMPs

In general, living organisms produce gene-encoded AMPs that provide an immediate defence mechanism upon invasion by pathogens (*Annunziato & Costantino, 2020*). However, the application of AMPs in the clinical setting has been retsricted due to pharmaceutical limitations such as poor bioavailability, susceptibility to enzymatic degradation and toxicity (*Deslouches et al., 2020*; *Costa et al., 2019*). For this reason, synthetic AMPs that can maintain therapeutic effectiveness with higher biological stability and a greater safety profile continue to be developed. Most synthetic AMPs are designed to recapture the amphiphilic properties of natural AMPs which are believed to be the primary determinants of their antibacterial activity. In other words, the natural peptides will be modified to produce *de novo* scaffolds that resemble the parent peptides (*Azmi, Skwarczynski & Toth, 2016*).

Historically, various methods were used to optimize AMPs from natural sources, thus generating synthetic variants. In 1881, the azide-coupling method was used by treating the silver salt of glycine with benzoylchloride to create the first N-protected dipeptide, benzoylglycylglycine. Nevertheless, in 1901, Emil Fischer reported the first synthetic dipeptides by hydrolysis of the glycine diketopiperazine, known as glycylglycine (*Lichtenthaler, 2002*). The development of temporary amino-protecting groups was required to overcome synthetic challenges. In 1931, the carbobenzoxy (Cbz) group was introduced, followed by the *ert*-butyloxycarbonyl (Boc) group in 1957 (*Anderson & McGregor, 1957*; *Grapsas, Cho & Mobashery, 1994*). The discovery of solid phase peptide synthesis (SPSS) was achieved in 1963, when peptide sequences were synthesized on solid support (*Merrifield, 1963*). However, the major limitations of SPSS included insufficient coupling and degradation reactions, as well as a build-up of by-products (*Pedersen et*

*al., 2012*; *Schnölzer et al., 2009*; *King, Fields & Fields, 2009*). For that reason, several new techniques of protein synthesis have been developed to overcome the limitations of SPPS. For example, chemical ligation and coupling two peptide fragments together were introduced (*Kemp & Kerkman, 1981*). Other ligation methods such as native chemical ligation (NCL), expressed protein ligation (EPL), and Staudinger ligation were also introduced to overcome the constraint (*Dawson et al., 1994*).

To date, methods for producing synthetic AMPs are constantly being improved. For example, using cationic peptides based on natural templates is becoming one of the most exciting new strategies for synthesizing AMPs today. These AMPs have complex mechanisms of action and do not readily lead to resistance. With their anti-inflammatory properties, as well as antimicrobial synergy, they hold promises as adjunctive strategies to supplement and enhance current therapies (*Fjell et al., 2012*). Several computational tools have also evolved in the development of more economical and powerful synthetic AMPs (*Cardoso et al., 2019*). For instances, the empirical methods, machine learning and *de novo* computational methods are being used in the optimization of peptides through random processes (*Porto, Silva & Franco, 2012*). Genetic algorithms also offer an alternative in the development of synthetic AMPs by identifying antibacterial activity-conferring determinants through successive generations of mutations and deletions in the target sequence (*Kliger, 2010*; *Fjell et al., 2011*). These candidates are refined over time as lower fitness values are removed from the candidate sequences (*Fjell et al., 2011*).

Hundreds of synthetic AMPs have been produced with the help of computer-aided design (*Wimley, 2019*). Previously, naturally occuring AMPs are used as templates to optimize their activity and stability by mutating one or more amino acid residues; this was followed by the *de novo* design of a variety of synthetic peptides, peptoids, peptidomimetics, oligomers and polymers (*Jiang et al., 2021*). An example is iseganan, where protegerin is used as the template and one or more amino acid residues has been mutated to other proteinogenic L-amino acids to achieve antimicrobial activity against gram-negative and gram-positive bacteria (*Trotti et al., 2004*). There are several other examples of synthetic AMPs that are produced using this approach, such as omiganan, which is developed using indolicin; and pexiganan,which in turn, is developed by magainin 2 (*Ge et al., 1999*; *Gottler & Ramamoorthy, 2009*; *Sader et al., 2004*).

Another way of producing synthetic AMPs is utilization of β-amino acids as the building blocks or using non-natural N-substituted amino acids (*Jiang et al., 2021*). For example, synthesized helical β-peptide that was developed from β-amino acid; and synthesized oligo-N-substituted-glycine-based helical peptoid that was developed by magainin 2 amide. Both of these synthetic AMPs show greater and more stable antibacterial activity compared to naturally occuring AMPs (*Chongsiriwatana et al., 2008*; *Patch & Barron, 2003*; *Cheng, Gellman & De Grado, 2001*). Generally, synthetic AMPs are more stable and possess better activity and selectivity compared to naturally occuring AMPs. However, the limitations of producing synthetic AMPs include the extended time required to do so and the high cost (*Jiang et al., 2021*).

## THE STRUCTURES OF AMPS

To clearly identify the potential of AMPs, the structure of AMPs needs to be well elucidated. AMPs are relatively short molecules, containing 12–100 amino acids with an amphipathic structure (*Hodges et al., 2011*). Several databases exist that manage information and conduct peptide analysis, due to the high numbers of natural, semi-synthetic and synthetic AMPs (*Mahlapuu et al., 2016*). AMPs can be classified based on their structure, amino acid composition and size. The structural features of AMPs can be divided into four main groups, (a) peptides with amphipathic α-helices (b) β sheets, (c) combined α-helices and β sheet structures (α β) known as a mixed structure and (d) non–α β structure known as extended structure (*Fig. 2*).

α-helical peptides are the most widely studied types of AMPs to date. The α-helical peptide has two amino acids adjacent to each other with a distance of 0.15 nm between them; the centre is about 100 degrees from the top view. Among the well-known peptides studied in this group are LL-37 and human lactoferricin (*Epand & Vogel, 1999*; *Hunter et al., 2005*; *Legrand et al., 2005*; *Pasupuleti, Schmidtchen & Malmsten, 2011*). In addition, among the other widely studied AMPs are colistin, melittin, nisin, and Cecropin A-Magainin 2 (CAMA) (*Bechinger & Lohner, 2006*; *Kumar, Kizhakkedathu & Straus, 2018*).

β-sheet peptides are composed of at least two β strands with disulfide bonds between these sheets. Interestingly, almost all β-sheet AMPs contain preserved cysteine residues and form disulphide bonds such as gomesin, polyphemusin, protegerin and tachyplesin (*Kumar, Kizhakkedathu & Straus, 2018*). However, some studies have reported short β-sheet forming AMPs that do not have disulfide bonds (*Cândido et al., 2019*; *Ong et al., 2014*; *Ong, Gao & Yang, 2013*). For example, the synthetic β-sheet AMP known as IK8-all D (irikirik-NH$_2$) which is derived from β-sheet forming peptides (IRIK)$_2$-NH$_2$ (IK8-all L) has shown no formation of disulfide bonds in its design (*Ong et al., 2014*).

Besides α-helical peptides and β-sheet peptides, there is a kind of AMP structure that had been found with the formation of α-helices and β-sheets (α β) (mixed structure). In this class of AMPs, the two monomers are packed against each other with the β-sheet of one monomer facing the α-helix of another monomer (*Kovaleva et al., 2020*). Human β-defensin-2 and pine defensin 1 (PsDef1) are among the peptides studied in this group (*Jenssen, Hamill & Hancock, 2006*; *Kovaleva et al., 2020*).

Extended/random coil AMPs display another unique structure that has been frequently discussed. This structure consists of two or more proline residues, tryptophan, arginine and histidine which have the capabilities to break the secondary structure elements (*Bahar & Ren, 2013*). In addition, many peptides such as indolicin and moricin, adopt their active structure only after they interact with the target cell membrane. Indolicin is a hemolytic AMP isolated from bovine neutrophils. It is effective as an antimicrobial agent because it has 13 tridecapeptide amides and an extremely high tryptophan content (*Cardoso et al., 2019*). Indolicin changes its structural profile to a ''boat-like''and transmembrane orientation to translocate the bacterial membrane and act on DNA (*Cardoso et al., 2019*). Moricin is a random coil AMP that was isolated from *Manduca sexta*. It consists of one aspartic acid,

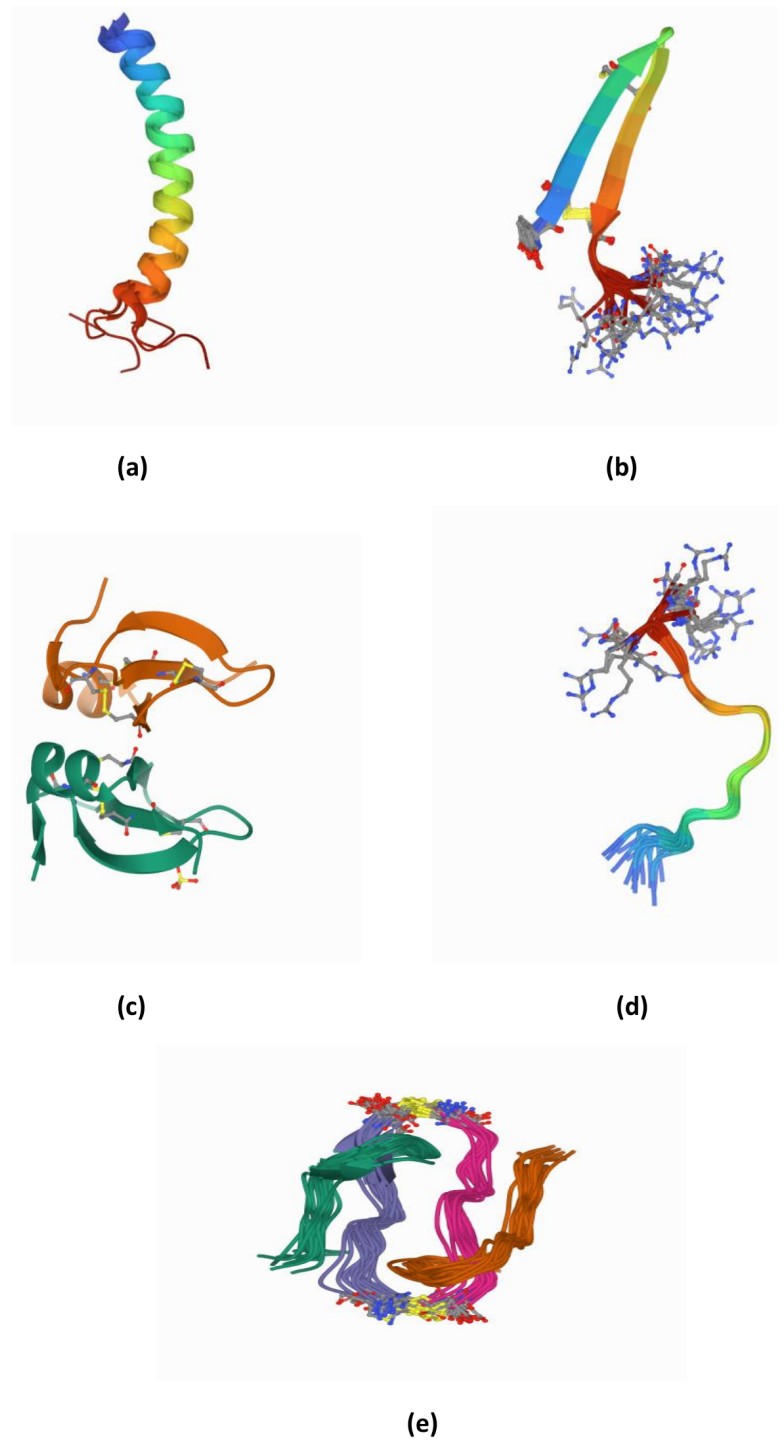

**Figure 2** (A–E) Structure of AMPs.

two arginine and nine lysine residues and features α-helical structures to perform their membrane-associated or intracellular mechanisms of action (*Dai et al., 2008*)

Apart from those, there have been progressively increasing reports in newly discovered AMPs with cyclic and disulfide-rich AMPs (Figure 2(e)) as well as AMPs with more complex topologies in the past two decades. Some studies reported these as the fifth class of AMPs (*Koehbach & Craik, 2019*). These peptides have been identified based on the nature of the peptide's cyclic topology such as "head to tail" or "head to side chain" as well as the nature of the crosslinks such as the presence of disulfide or thioether bridges (*Koehbach & Craik, 2019*). For example, microcin J25 (lasso peptides) has been found with a head-to-side chain cycle (magenta) threaded by the C-terminal tail and sterically locked in place by bulky residues (cyan) (*Rosengren et al., 2003*).

## THE BACTERIAL RESISTANCE MECHANISMS TOWARDS ANTIBIOTICS AND THE WAY AMPS CAN HELP

Understanding on the mechanisms of antibiotic resistance will allow the relevance of AMPs to be seen as a potential alternative for antibiotics. Antibiotic resistance mechanisms in bacteria and how AMP mechanisms aid in killing bacteria are depicted in Fig. 3. This figure displays four major molecular mechanisms by which bacteria can withstand antibiotic effects. Among these are drug-target modifications, antibiotic-degrading enzymes, antibiotic-altering enzymes and antibiotic efflux pumps (*Laws, Shaaban & Rahman, 2019*). These resistance mechanisms can occur in one bacterial cell simultaneously, resulting in high levels of resistance to various antibiotic compounds (*Peterson & Kaur, 2018*). In addition to these four major mechanisms, bacterial biofilm has also attracted a great attention in resistance mechanisms towards antibiotics. Bacteria that attach to the surface and grow as biofilm are protected from killing by antibiotics, thus makes the treatment difficult (*Dincer, Uslu & Delik, 2020*).

Meanwhile, most of the AMPs were found to kill bacterial cells by disrupting the bilayer membrane without the interference of all the available antibiotic resistance characters that might be present in a bacterial cell. However, there are studies indicating that bacteria can resist AMPs treatment at sub-lethal doses and expel them by efficient efflux pumps (*Cardoso et al., 2017*). Membrane interactions are important in the direct antimicrobial activity of AMPs (*Hollmann et al., 2018*; *Lei et al., 2019*). Several models have been introduced to explain the mechanism of disrupting bilayer membranes by AMPs. These models are the barrel-stave model, the toroidal-pore model, and the carpet model. In the barrel-stave model, recruitment of additional peptides placed perpendicularly into the bilayer will lead to the formation of a peptide-lined transmembrane pore. In this pore, the peptides align with the hydrophobic side facing the lipid core of the membrane, while the hydrophilic regions face the interior region of the pore. In the toroidal-pore model, phospholipids bend continuously from one leaflet to another due to the interaction of AMPs. This then results in a pore lined by both peptides and the head groups of phospholipids. For the carpet model, the mechanism is explained by the formation of micelles due to membrane

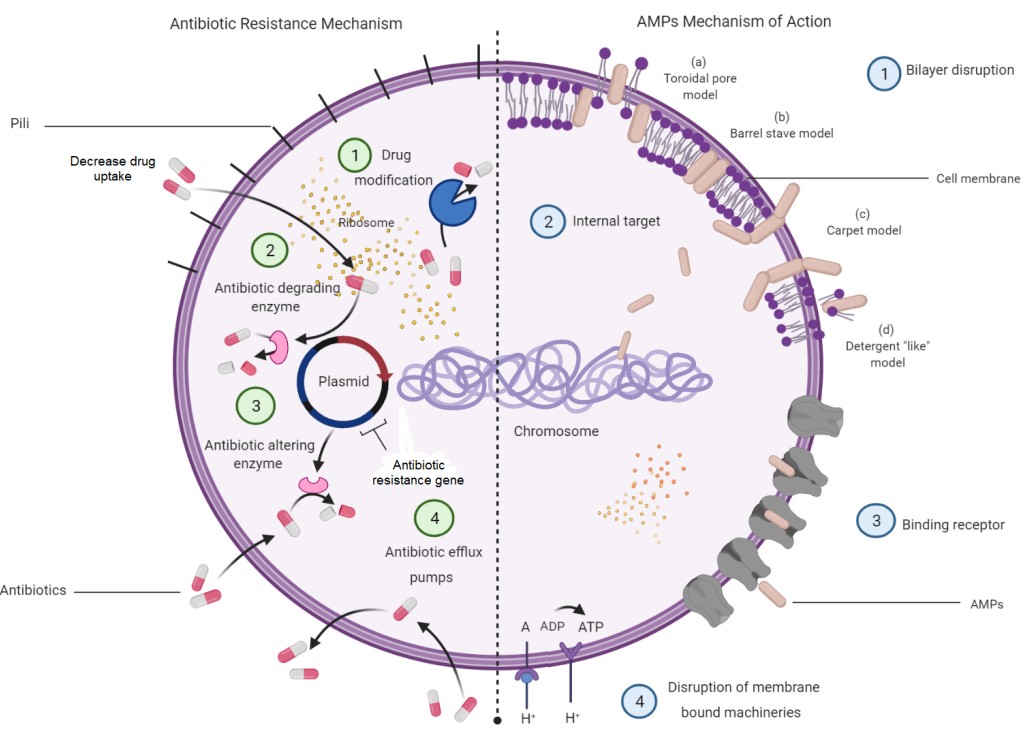

**Figure 3 Bacterial resistance mechanisms to antibiotics and the mechanisms of AMPs in bacteria.**

disruptions by the tension in the bilayer as a result of peptide accumulation (*Mahlapuu et al., 2016*).

Intracellular targeting and inhibition of protein synthesis also act as targets by which some of the AMPs may interfere and express their function and ability to disrupt the cell growth. To reach the cytoplasmic membrane of Gram-negative bacteria, AMPs will translocate through the outer membrane *via* a self-promoted uptake (*Le, Fang & Sekaran, 2017*; *Mahlapuu et al., 2016*). Moreover, activated AMPs cause damages to bacterial cells by attacking an internal target or translocating across the membrane receptors, entering the bacterial cytoplasm and disrupting intracellular targets (*Jindal et al., 2015*; *Malanovic & Lohner, 2016*). Bacterial destruction also occurs by the interaction between the electrostatic forces of the positively charged amino acids of the AMPs and the negatively charged cell surface. These create an ion-permeable channel and increase membrane permeability to develop cleavages (*Lin & Weibel, 2016*).

The translocation of AMPs will not only disrupt the cellular membrane but also target some important processes, such as DNA transcription and replication, RNA synthesis and protein synthesis, enzymatic activity and protein folding or cell wall synthesis (*Le, Fang & Sekaran, 2017*). For example, indolicin acts by targeting DNA and inhibiting the replication process, indirectly killing the bacteria. Bacterial death caused by AMPs could be the result of multiple and complementary actions. The mode of action of AMPs depends on several factors, including peptide concentrations, the targeted bacterial species, tissue localization

and the bacterial growth phase (*Kumar, Kizhakkedathu & Straus, 2018*; *Mahlapuu et al., 2016*).

Bacterial cytoplasmic membranes are rich with negatively charged phospholipids, including phosphatidylglycerol, cardiolipin and phosphatidylserine, all of which are highly attracted to the positive charges of AMPs (*Ebenhan et al., 2014*). Gram-negative bacteria consist of an additional lipopolysaccharide-rich outer membrane that acts as a barrier to the cytoplasmic membranes. The presence of teichoic acids in the cell wall of Gram-positive bacteria also provides an additional electronegative charge to the bacterial surface (*Ebenhan et al., 2014*). As opposed to bacteria, human cells seem to be rich in neutrally charged phospholipids, such as phosphatidylethanolamine, phosphatidylcholine and sphingomyelin. This fundamental difference between microbial and mammalian membranes has made AMPs a highly selective agent against bacteria (*Ebenhan et al., 2014*). The presence of cholesterol in humans affects the fluidity of the phospholipid in the membranes *via* an increased stability of the bilayer, then, reduces the activity of AMPs *via* stabilization of phospholipids bilayer (*Subczynski et al., 2018*).

## THE ADVANTAGES AND DISADVANTAGES OF AMPS

In developing AMPs as the potential treatment for antibiotic-resistant pathogens, the advantages as well as the limitations of AMPs should be considered and are discussed in this topic.

### Advantages of AMPs

The long-term and overly frequent use of conventional antibiotics as antibacterial agents has the potential to cause mutations in the bacterium, thereby increasing resistance to the antibiotics themselves (*Bahar & Ren, 2013*). This issue has prompted researchers and the pharmaceutical industry to focus on identifying drugs capable of replacing antibiotics. AMPs are a type of cationic peptide, an agent thought to be able to fulfil the role of antibiotics. Unlike antibiotics, AMPs interact with the cell membrane of bacteria by neutralizing the charge and, subsequently, causing bacterial death by penetrating the membrane, thereby reducing the risk of bacterial resistance (*Mahlapuu et al., 2016*). This ability on the part of AMPs indicates them to be more effective than conventional antibiotics.

AMPs have demonstrated a wide range of capabilities in killing bacteria as well as fungi and viruses (*Amso & Hayouka, 2019*; *Mahlapuu et al., 2016*). Interestingly, AMPs have less side effects on the hosts, as their uses cause a very minimal toxicity to the body based on previous studies (*Zharkova et al., 2019*; *Lei et al., 2019*; *Mahlapuu et al., 2016*). For example, a peptide known as citrus-amp1, which is isolated from citrus, exhibited low toxicity effects when tested on *Galleria mellonella*, a cell line derived from the larval-fat body tissues of the wax moth, and on U87 MG, a human glioblastoma cell line commonly used as a model for cytotoxicity (*Kishi et al., 2018*). A peptide known as Nisin A also presented low toxicity effects when tested on HT29 and Caco-2 cells by using MTT assay (*Maher & McClean, 2006*).

AMPs with a simple structure–activity relationship are widely used in the development of medicines. They are particularly useful in this regard because they are associated with excellent water stability and solubility (*Dehsorkhi, Castelletto & Hamley, 2014*). For example, daptomycin, another type of AMPs, has been used as an anionic antibacterial peptide to treat skin infections stemming from Gram-positive bacteria, thereby showing inhibitory effects on *S. aureus* and typhoid bacillus *Salmonella typhi* (*Lei et al., 2019*).

Additionally, AMPs have also demonstrated a good inhibition of cancer cells (*Mahlapuu et al., 2016*). In fact, cancer cells are more sensitive to AMPs than normal cells. This is because the cytoskeletons of cancer cells do not grow well when compared with those of normal cells, which allows AMPs to easily enter the lipid membrane and form ion channels or pores. This process eventually destroys the cancer cells by causing the leakage of the cell content (*Jäkel et al., 2012*; *Mahlapuu et al., 2016*). More specifically, the content of those cationic AMPs associated with the high acid phospholipids that occupy the outer surface of cancerous cell causes changes in the membrane, extracellular matrix and cytoskeleton (*Mahlapuu et al., 2016*). The loss of phospholipids asymmetry in cancer cells provides them with more negatively charged residues in their upper leaflet, thus favouring electrostatic attraction of AMPs (*Ramos-Martín & D'Amelio, 2021*). In terms of acting as antimicrobial agents, AMPs have the potential to fight antibiotic-resistant bacteria. The bactericidal effect of AMPs is generally due to the creation of pores in the bacterial cytoplasmic membrane, which results in a loss of control over the flow of ions through the membrane and, consequently, cell deaths. This renders the use of AMPs a promising strategy for addressing the problem of antibiotic resistance through fulfilling the role of conventional antibiotics (*Lei et al., 2019*).

## Disadvantages of AMPs

Despite the uniqueness and recognised advantages of AMPs, concerns have been raised about certain disadvantages of their excessive use that may eventually lead to the emergence of resistance against AMPs as bacteria will always mutate for survival. Among other disadvantages include several aspects such as toxicity, immunogenicity, haemolytic activity in certain type of human cells, reduced activity based on salt sensitivity, and the high cost of production (*Aoki & Ueda, 2013*; *Moravej et al., 2018*). These characteristics render the use of AMPs in the field of medicine more difficult.

There have been challenges in classifying the good AMPs and AMPs that can cause side effects. In some cases, the use of AMPs is associated with a high risk of toxic effects in human cells. For example, certain peptides such as arenicin, LTX-109 and LL-37 have been found to cause side effects (itching, burning and pain) to mammalian cells *in vitro* and, further, to be toxic with the formation of pore at the membrane, disruption of the membrane and cell lysis, when injected into the bloodstream (*Patrulea, Borchard & Jordan, 2020*). This problem urges research to look for more new AMPs compounds with less toxicity effects. In addition, although AMPs have been reported not to elicit an immunogenic response (no interference from the action of the host cell), immunogenicity continues to be a concern and even a serious problem in the development of the peptide drugs (*Mahlapuu et al., 2016*). Based on previous findings, structural properties such as the changes in peptide

sequences (modified amino acids), glycosylation changes, the presence of aggregates and other possible factors have been identified as the factors that can lead to immunogenicity of AMPs (*da Cunha et al., 2017*; *Natalia, Brendan & Sam., 2017*). These factors may cause the function of AMPs to be disrupted.

A certain number of AMPs have been reported to influence haemolytic activity. Indolicidin, for example, a 13-residue cationic peptide that is rich in tryptophan, has been found to exhibit a broad spectrum of anti-bacterial activity, however it exhibits haemolytic activity that limits their clinical applications (*Mirski et al., 2018*). Some types of AMPs can interact directly with the host cell and dissolve it, although most AMPs bind to the bacterial opening through electrostatic interactions. The amide peptides exhibit higher antimicrobial activity than natural AMPs, although they are more haemolytic. In addition, the functional analysis of AMPs has demonstrated how their high amphiphilicity and high hydrophobicity contribute to their increased haemolytic capability (*Aoki & Ueda, 2013*; *Bahar & Ren, 2013*). However, the haemolytic activity of several AMPs was observed to be different in certain types of different species. For example, based on a previous study, 24 AMPs were evaluated for their haemolytic activity in cells of four different species such as human, dog, rat and bovine. Based on this study, some of the AMPs showed no or less haemolytic activity towards each species, and vice versa (*Greco et al., 2020*). More thorough studies need to be conducted to identify the most appropriate AMPs that do not cause harm to human.

The fact that some AMPs require electrostatic interactions with microbial membranes to form a skeletal structure has caused them to be more sensitive to salt, which often leads to problems with clinical applications (*Andersson, Hughes & Kubicek-Sutherland, 2016*; *Hollman et al., 2018*). Human body fluids that have a high salt concentration disrupt the function of these AMPs and, therefore, deactivate them (*Bastos et al., 2018*). Thus, the identification of salt-resistant AMPs is essential to improve the effectiveness of AMPs within the human body. AMPs are also rapidly degraded in human body by proteases (*Aoki & Ueda, 2013*). A feasible production method is required to develop AMPs as drugs. Further studies need to be conducted in this regard, and such efforts will require a lot of investments. For instance, the production of heterologous AMPs within prokaryotic systems is considered to be extremely difficult, as AMPs are associated with the poisoning risk in prokaryotic cells (*Aoki & Ueda, 2013*).

## CONCLUSION

Physicians and scientists describe the antibiotic resistance crisis as increasingly threatening. This problem may worsen unless the public takes precautionary measures. This issue has prompted research to look for new antimicrobials, leading to ongoing research on the AMPs. However, the development has been slow due to several challenges such as high cost of salvage, potential toxicity and lack of solid guideline for rational design. In addition, it might be possible to modify the characteristics of naturally occurring AMPs to produce synthetic AMPs through certain available methods, but to predict the impact of these modifications in clinical usage is still challenging. Thus, there is a need to observe

the effects of structural modifications on their function, activity, and target spectrum. On the other hand, the mechanisms of action of AMPs targeting the cell membrane, make them as a good potential approach in controlling the antibiotic resistant pathogen. A close collaboration between different disciplines and the development of new tools that can decipher the structure-function relationship, as well as efficiently synthesize and modify AMPs molecules will be the key to AMPs related research in the future.

**Abbreviations**

| | |
|---|---|
| **AMPs** | Antimicrobial peptides |
| **GRE** | Gentamicin-Resistance *Enterecoccus* |
| **MRSA** | Methicillin-Resistant *Staphylococcus Aureus* |
| **MR-CoNS** | Methicillin resistant coagulase-negative *Staphylococci* |
| **HGT** | horizontal gene transfer |
| **CAMA** | Cecropin A-Magainin 2 |
| **DNA** | Deoxyribonucleic acid |
| **RNA** | Ribonucleic acid |
| **U87 MG** | Uppsala 87 Malignant Glioma |
| **LL-37** | form of LL-37 |
| ***X. laevis*** | *(Xenopus laevis)* |
| **SPSS** | solid phase peptide synthesis |
| **NCL** | native chemical ligation |
| **EPL** | expressed protein ligation |

### Funding
This study was supported by the funds of Ministry of Higher Education, Malaysia and Universiti Putra Malaysia through Fundamental Research Grant Scheme (FRGS/1/2017/SKK11/UPM/01/1) and Putra Grant (GP/2017/9571800). The funders had no role in study design, data collection and analysis, decision to publish, or preparation of the manuscript.

### Grant Disclosures
The following grant information was disclosed by the authors:
The funds of Ministry of Higher Education, Malaysia and Universiti Putra Malaysia through Fundamental Research Grant Scheme: FRGS/1/2017/SKK11/UPM/01/1.
Putra Grant: GP/2017/9571800.

### Competing Interests
The authors declare there are no competing interests.

## Author Contributions

- Nurul Hana Zainal Baharin and Mohd Nasir Mohd Desa conceived and designed the experiments, performed the experiments, analyzed the data, prepared figures and/or tables, authored or reviewed drafts of the paper, and approved the final draft.
- Nur Fadhilah Khairil Mokhtar conceived and designed the experiments, performed the experiments, analyzed the data, authored or reviewed drafts of the paper, and approved the final draft.
- Banulata Gopalsamy conceived and designed the experiments, analyzed the data, authored or reviewed drafts of the paper, and approved the final draft.
- Nor Nadiha Mohd Zaki, Mohd Hafis Yuswan, Sahar Abbasiliasi, Amalia Mohd Hashim, Muhamad Shirwan Abdullah Sani and Shuhaimi Mustafa performed the experiments, analyzed the data, authored or reviewed drafts of the paper, and approved the final draft.
- AbdulRahman Muthanna and Nurul Diana Dzaraly conceived and designed the experiments, analyzed the data, prepared figures and/or tables, authored or reviewed drafts of the paper, and approved the final draft.

## Data Availability

This is a literature review study.

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
