# Peer review of "The characteristics and roles of antimicrobial peptides as potential treatment for antibiotic-resistant pathogens: a review"

_PeerJ, doi:10.7717/peerj.12193_

## Round 0.1 · original submission · Major Revisions

Dear Dr. Baharin and colleagues:

Thanks for submitting your manuscript to PeerJ. I have now received two independent reviews of your work, and as you will see, the reviewers raised some concerns about the review. Despite this, these reviewers are optimistic about your work and the potential impact it will have on research studying antimicrobial peptides as therapeutics treating for infectious by antibiotic-resistant pathogens. Thus, I encourage you to revise your manuscript, accordingly, taking into account all of the concerns raised by both reviewers.

Please ensure that an English expert has edited your revised manuscript for content and clarity. Please also ensure that your figures and tables contain all of the information that is necessary to support your findings and observations.

There are many comments by both reviewers that ask for more information on specific issues; please address these.

I look forward to seeing your revision, and thanks again for submitting your work to PeerJ.

Good luck with your revision,

-joe

·

Basic reporting

In the review article submitted by Baharin and co-workers under the title “The characteristics and roles of antimicrobial peptides as potential treatment for antibiotic-resistant pathogens: A review”, the authors discuss the characteristics, advantages and disadvantages of antimicrobial peptides for their potential in treating antibiotic-resistant pathogens. The topic is important and of great interest to a good section of researchers in the medical/pharmaceutical field.

- Multiple recent review articles have been published addressing a similar topic, the authors need to highlight what is new and different from what has been recently reviewed in the literature as in the following examples:
o Dijksteel GS, Ulrich MMW, Middelkoop E, Boekema BKHL. Review: Lessons Learned From Clinical Trials Using Antimicrobial Peptides (AMPs). Front Microbiol. 2021 Feb 22;12:616979. doi: 10.3389/fmicb.2021.616979. PMID: 33692766; PMCID: PMC7937881.

o Ryu M, Park J, Yeom JH, Joo M, Lee K. Rediscovery of antimicrobial peptides as therapeutic agents. J Microbiol. 2021 Feb;59(2):113-123. doi: 10.1007/s12275-021-0649-z. Epub 2021 Feb 1. PMID: 33527313.

- The authors talked extensively about the dilemma of antibiotics; this can be shortened as this is not the focus of the review. Instead, the authors can highlight the impact of antibiotics resistance on mortality, prognosis, rise in the cost of care, etc.) by providing data with documented references.

Experimental design

-In the methodology, please indicate if the assessment of the included articles was performed by multiple individuals for the same article(s) or each individual was assigned alone to certain articles to be assessed, especially that the number of the authors is unusually large.

Validity of the findings

- The comparison in table 1 is somehow oversimplifying the differences generalizing the features of some AMPs overall antibiotics based on findings in a very limited number of studies.

Additional comments

- The sentences in the review in many instances tend to be very long; three lines and even longer, the authors can consider shortening some of them for more clarity.
- In the abstract, the authors abruptly mentioned sepsis treatment by novel AMPs, why it was singled out away from other types of infections?
- Data in figure 1 is outdated and needs to be updated.
- The quality of both figures 2 and 3 needs to be improved and their source needs to acknowledge if they are not the original work of the authors. If the latter is the case, then they need to cite in the legend how they were generated.
- Only two references are from 2020, more recent findings need to be included.
Minor comments:
Line 87: resistance should be resistant
Line 87: Staphylococci should not be italicized
Line 202: CAMA should be mentioned in full first before being abbreviated
The titles of the subsections are uniformly written regarding the capitalization of the words in the title
Line 307: Staphylococcus aureus should be abbreviated
Line 307: “Typhoid bacillus” is this the name of a bacterial species?

Reviewer 2 ·

Basic reporting

The manuscript by Baharin et al. offers an overview of the current multi-drug resistant bacterial infection crisis and how antimicrobial peptide could be used as a promising therapeutic strategy. The manuscript is divided into subsections that help the reader to follow the authors’ ideas. Nevertheless, I have minor and major concerns about the manuscript. It is not evident how this review article differs from hundreds of other review articles regarding AMPs in the context of resistant bacterial infections. Moreover, in the end of the Introduction, the authors claim that the paper provides an overview on the synthetic and long-lasting AMPs analogues. However, this was not accomplished in any part of the manuscript, as the authors described (with lack of details) many naturally occurring AMPs, instead of focusing on optimized AMPs and their improved characteristics over their parent peptides. Some figures should be reorganized. Please find below a list of comments that should be addressed:

Line 49 and 51: The authors start with “Today, the number of deaths…” However, they cite a reference from 2018. The authors should consider updating this for a more recent reference.

Please use either “anti-microbial peptides” or “antimicrobial peptides”. Both forms appear in the main text.

Line 57: The authors affirm “They are host defense peptides…” Please rephrase, not all AMP are HDPs. For instance, synthetic computationally designed peptides and peptidomimetics with improved therapeutic potential are not HDPs.

Line 57 – 59. The authors mention only membrane-associated mechanisms of action. It would be interesting to add a short sentence explaining that some AMPs can also translocate across bacterial membranes to act on intracellular targets, including ribosomes. Non-lytic AMPs may include, buforin II, indolicidin, drosocin, among others.

A native English speaker should revise the manuscript. There are some minor issues.

In some sections of the manuscript, there are repetitive statements. For instance “the overuse and inappropriate prescribing of antibiotics has led to the emergence of more resistant bacteria and toxicity problems”. Although this information is relevant, the authors should carefully check the entire manuscript to modulate the use of repetitive sentences.

This work is focused on AMPs against bacterial infections. It was quite strange for me to see that the terms “germs” and “germs infections” were overused in the manuscript. These terms are too broad in my view and the authors should consider being more specific (e.g., bacterial, fungi, virus, etc.).

Lines 130 – 131: “The less developed body’s immune defence may then increase infection risk in a case of infectious germ transmission”. Very confusing sentence, please rephrase.

Lines 133 to 136 lack literature citations. Please add.

Figure 1: Why this particular database? When I clicked to check the database it appears: server not found. Considering that the authors were very clear on how they used Google Scholar, Pubmed, etc, to find the most relevant articles for their review, I also encourage them to better organize the data in this figure. Initially, the graph must be edited. For instance, the y axis says “percentage”. But percentage of what? AMP sequences? What is “percentage of diversity” in this case? It is not clear. Moreover, this analysis should be highly improved by comparing different databases, including APD, YAMPD, CAMP among others. I could not find the citation (Mahlapuu et al., 2019) described in the figure 1 caption in the references list. Please double-check.

Avoid expressions like “good antibacterial activity (line 139)”, “good” is too relative.

In the topic “The Archival and the Diversity of AMPs” the authors focused only on naturally occurring AMPs. It should be in the topic title. Actually, the authors affirm in the end of the Introduction “This paper provides an overview on the synthetic and long-lasting AMPs analogues”, but little is really described and discussed about synthetic peptides... Therefore, at some points, the manuscript lacks focus considering the authors’ goals.

Line 193 – 195: Same as comment 1.

Figure 2 must be completely redesigned. The quality is poor. There is not description of the PDB IDs used in the figure. Why did the authors bind to an ion? Why did the authors not specify disulfide bonds? Moreover, this figure can be highly improved if the authors add more AMP structural scaffolds (e.g., cyclotides and other head-to-tail or side-to-side chain cyclic peptides, stapled peptides, among others.)

Line 201: What did the authors mean by rolling indolicidin? I understand that some chemical modifications may favor helical structure in indolicidin-like peptides. However, this is a well-known AMP that cause non-lytic effects on bacterial and presents random coil arrangements… There are many other well-known AMPs that adopt α-helix that would be better examples.

Line 202. The same as above… Defensin is a mix of helix, loops and beta-sheet. Not the best example for α-helical AMPs. The authors should rethink their examples in this entire section.

Line 204. Some studies have reported short β-sheet forming AMPs that do not present disulfide bonds (please check: 10.1002/adfm.201202850, 10.1021/acsinfecdis.9b00073, 10.1016/j.biomaterials.2013.10.053).

Line 206. Same of line 2. It would be better if the authors opened a subsection for mixed structures (e.g, defensins).

Line 216: “Besides, many peptides form their active structure only after” please, replace “form” by “adopt”. Moreover, the authors use as example, indolicidin. Indeed, this AMP changes its structural profile to a “boat-like” structural orientation to translocate bacterial membrane and act on DNA. However, it is not clear in the present manuscript that such structural transition occurs. For someone that it not an expert in AMPs the best examples, by far, would be random coil AMPs in solution that, when in contact with bacterial surfaces, adopt α-helical structures to perform their membrane-associated or intracellular mechanisms of action.

In general the “AMP structure” section should be highly improved. What about cyclic peptides? Different structural scaffolds? And, once again, considering the last sentence of the introduction, I believe the authors should provide the readers with more examples for optimized AMPs, not only naturally occurring ones.

Figure 3 looks much better. A more detailed caption should be organized for this one, describing (very briefly) the mechanisms.

“The Bacterial Resistance Mechanisms towards Antibiotics and the Way AMPs can help” How is this topic different from hundreds of other review articles regarding AMPs? Moreover, some of the statements are controversial. There are studies indicating that bacterial can resist to AMP treatment at sub-lethal doses. Most changes occur in the bacteria surface, efflux pumps production, among others (10.1038/s41598-017-04181-y). Finally, bacterial biofilms have attracted great attention in the last five years. Why did the authors decide not to mention this resistance mechanism, which is currently responsible for >70% of all bacterial infections in humans?

“Advantages of AMPs”: the authors mention AMP synergism with antibiotics as an advantage. Please highlight that such strategy is highly controversial. We are still to fully understand the determinants for AMP/antibiotics synergism and, because of that, some results are positive, some are not, some are effective in vitro, but not in vivo and vice-versa. Therefore, maybe the authors should not emphasize this as an advantage.

“Interestingly, AMPs have no/less side effects for the user, as their use causes zero to only very minimal toxicity to the body when compared with other antibiotics”. This sentence is very confusing, please rephrase. Moreover, the authors give only on example of a peptide “citrus-amp1” with non-toxic effects. Please, add more examples.

“AMPs, which have a low synthetic cost, are small molecules that are” please review this statement carefully. Low synthesis cost compare to what? Chemically modified AMPs (D-amino acids, for instance) can be quite expensive. Moreover, they are more expensive to produce then most antimicrobials already available in the market and, this characteristic has been highlighted as a bottleneck in AMP-based therapies. Finally, small molecules are usually describe with up to 200 Da, which is not the case here (AMPs).

The explanation about the anti-cancer activities of AMPs is not complete…. The authors should also mention that the phospholipids asymmetry in cancer cells provides them with more negatively charged residues in their upper leaflet, thus favoring electrostatic attraction of cationic AMPs.

Lines 365 – 367: “Moreover, the cost of production of AMPs is very high when compared with the production costs of conventional antibiotics” The authors should carefully review the manuscript, as they are often controversial.

The relevance of Table 1 is questionable. Antibiotics and AMPs are highly different classes of antimicrobials. If we check carefully, most studies not even consider their correlation in number of mols for biological experiments. I do believe that AMPs represent a promising strategy, but it is not like they will completely replace antibiotics in a short-term. A more interesting comparison here, would be between naturally occurring and optimized AMPs (which should be the focus of this manuscript according with its title and last sentence of the Introduction).

Experimental design

The manuscript by Baharin et al. offers an overview of the current multi-drug resistant bacterial infection crisis and how antimicrobial peptide could be used as a promising therapeutic strategy. The manuscript is divided into subsections that help the reader to follow the authors’ ideas. Nevertheless, I have minor and major concerns about the manuscript. It is not evident how this review article differs from hundreds of other review articles regarding AMPs in the context of resistant bacterial infections. Moreover, in the end of the Introduction, the authors claim that the paper provides an overview on the synthetic and long-lasting AMPs analogues. However, this was not accomplished in any part of the manuscript, as the authors described (with lack of details) many naturally occurring AMPs, instead of focusing on optimized AMPs and their improved characteristics over their parent peptides. Some figures should be reorganized. Please find below a list of comments that should be addressed:

Line 49 and 51: The authors start with “Today, the number of deaths…” However, they cite a reference from 2018. The authors should consider updating this for a more recent reference.

Please use either “anti-microbial peptides” or “antimicrobial peptides”. Both forms appear in the main text.

Line 57: The authors affirm “They are host defense peptides…” Please rephrase, not all AMP are HDPs. For instance, synthetic computationally designed peptides and peptidomimetics with improved therapeutic potential are not HDPs.

Line 57 – 59. The authors mention only membrane-associated mechanisms of action. It would be interesting to add a short sentence explaining that some AMPs can also translocate across bacterial membranes to act on intracellular targets, including ribosomes. Non-lytic AMPs may include, buforin II, indolicidin, drosocin, among others.

A native English speaker should revise the manuscript. There are some minor issues.

In some sections of the manuscript, there are repetitive statements. For instance “the overuse and inappropriate prescribing of antibiotics has led to the emergence of more resistant bacteria and toxicity problems”. Although this information is relevant, the authors should carefully check the entire manuscript to modulate the use of repetitive sentences.

This work is focused on AMPs against bacterial infections. It was quite strange for me to see that the terms “germs” and “germs infections” were overused in the manuscript. These terms are too broad in my view and the authors should consider being more specific (e.g., bacterial, fungi, virus, etc.).

Lines 130 – 131: “The less developed body’s immune defence may then increase infection risk in a case of infectious germ transmission”. Very confusing sentence, please rephrase.

Lines 133 to 136 lack literature citations. Please add.

Figure 1: Why this particular database? When I clicked to check the database it appears: server not found. Considering that the authors were very clear on how they used Google Scholar, Pubmed, etc, to find the most relevant articles for their review, I also encourage them to better organize the data in this figure. Initially, the graph must be edited. For instance, the y axis says “percentage”. But percentage of what? AMP sequences? What is “percentage of diversity” in this case? It is not clear. Moreover, this analysis should be highly improved by comparing different databases, including APD, YAMPD, CAMP among others. I could not find the citation (Mahlapuu et al., 2019) described in the figure 1 caption in the references list. Please double-check.

Avoid expressions like “good antibacterial activity (line 139)”, “good” is too relative.

In the topic “The Archival and the Diversity of AMPs” the authors focused only on naturally occurring AMPs. It should be in the topic title. Actually, the authors affirm in the end of the Introduction “This paper provides an overview on the synthetic and long-lasting AMPs analogues”, but little is really described and discussed about synthetic peptides... Therefore, at some points, the manuscript lacks focus considering the authors’ goals.

Line 193 – 195: Same as comment 1.

Figure 2 must be completely redesigned. The quality is poor. There is not description of the PDB IDs used in the figure. Why did the authors bind to an ion? Why did the authors not specify disulfide bonds? Moreover, this figure can be highly improved if the authors add more AMP structural scaffolds (e.g., cyclotides and other head-to-tail or side-to-side chain cyclic peptides, stapled peptides, among others.)

Line 201: What did the authors mean by rolling indolicidin? I understand that some chemical modifications may favor helical structure in indolicidin-like peptides. However, this is a well-known AMP that cause non-lytic effects on bacterial and presents random coil arrangements… There are many other well-known AMPs that adopt α-helix that would be better examples.

Line 202. The same as above… Defensin is a mix of helix, loops and beta-sheet. Not the best example for α-helical AMPs. The authors should rethink their examples in this entire section.

Line 204. Some studies have reported short β-sheet forming AMPs that do not present disulfide bonds (please check: 10.1002/adfm.201202850, 10.1021/acsinfecdis.9b00073, 10.1016/j.biomaterials.2013.10.053).

Line 206. Same of line 2. It would be better if the authors opened a subsection for mixed structures (e.g, defensins).

Line 216: “Besides, many peptides form their active structure only after” please, replace “form” by “adopt”. Moreover, the authors use as example, indolicidin. Indeed, this AMP changes its structural profile to a “boat-like” structural orientation to translocate bacterial membrane and act on DNA. However, it is not clear in the present manuscript that such structural transition occurs. For someone that it not an expert in AMPs the best examples, by far, would be random coil AMPs in solution that, when in contact with bacterial surfaces, adopt α-helical structures to perform their membrane-associated or intracellular mechanisms of action.

In general the “AMP structure” section should be highly improved. What about cyclic peptides? Different structural scaffolds? And, once again, considering the last sentence of the introduction, I believe the authors should provide the readers with more examples for optimized AMPs, not only naturally occurring ones.

Figure 3 looks much better. A more detailed caption should be organized for this one, describing (very briefly) the mechanisms.

“The Bacterial Resistance Mechanisms towards Antibiotics and the Way AMPs can help” How is this topic different from hundreds of other review articles regarding AMPs? Moreover, some of the statements are controversial. There are studies indicating that bacterial can resist to AMP treatment at sub-lethal doses. Most changes occur in the bacteria surface, efflux pumps production, among others (10.1038/s41598-017-04181-y). Finally, bacterial biofilms have attracted great attention in the last five years. Why did the authors decide not to mention this resistance mechanism, which is currently responsible for >70% of all bacterial infections in humans?

“Advantages of AMPs”: the authors mention AMP synergism with antibiotics as an advantage. Please highlight that such strategy is highly controversial. We are still to fully understand the determinants for AMP/antibiotics synergism and, because of that, some results are positive, some are not, some are effective in vitro, but not in vivo and vice-versa. Therefore, maybe the authors should not emphasize this as an advantage.

“Interestingly, AMPs have no/less side effects for the user, as their use causes zero to only very minimal toxicity to the body when compared with other antibiotics”. This sentence is very confusing, please rephrase. Moreover, the authors give only on example of a peptide “citrus-amp1” with non-toxic effects. Please, add more examples.

“AMPs, which have a low synthetic cost, are small molecules that are” please review this statement carefully. Low synthesis cost compare to what? Chemically modified AMPs (D-amino acids, for instance) can be quite expensive. Moreover, they are more expensive to produce then most antimicrobials already available in the market and, this characteristic has been highlighted as a bottleneck in AMP-based therapies. Finally, small molecules are usually describe with up to 200 Da, which is not the case here (AMPs).

The explanation about the anti-cancer activities of AMPs is not complete…. The authors should also mention that the phospholipids asymmetry in cancer cells provides them with more negatively charged residues in their upper leaflet, thus favoring electrostatic attraction of cationic AMPs.

Lines 365 – 367: “Moreover, the cost of production of AMPs is very high when compared with the production costs of conventional antibiotics” The authors should carefully review the manuscript, as they are often controversial.

The relevance of Table 1 is questionable. Antibiotics and AMPs are highly different classes of antimicrobials. If we check carefully, most studies not even consider their correlation in number of mols for biological experiments. I do believe that AMPs represent a promising strategy, but it is not like they will completely replace antibiotics in a short-term. A more interesting comparison here, would be between naturally occurring and optimized AMPs (which should be the focus of this manuscript according with its title and last sentence of the Introduction).

Validity of the findings

The manuscript by Baharin et al. offers an overview of the current multi-drug resistant bacterial infection crisis and how antimicrobial peptide could be used as a promising therapeutic strategy. The manuscript is divided into subsections that help the reader to follow the authors’ ideas. Nevertheless, I have minor and major concerns about the manuscript. It is not evident how this review article differs from hundreds of other review articles regarding AMPs in the context of resistant bacterial infections. Moreover, in the end of the Introduction, the authors claim that the paper provides an overview on the synthetic and long-lasting AMPs analogues. However, this was not accomplished in any part of the manuscript, as the authors described (with lack of details) many naturally occurring AMPs, instead of focusing on optimized AMPs and their improved characteristics over their parent peptides. Some figures should be reorganized. Please find below a list of comments that should be addressed:

Line 49 and 51: The authors start with “Today, the number of deaths…” However, they cite a reference from 2018. The authors should consider updating this for a more recent reference.

Please use either “anti-microbial peptides” or “antimicrobial peptides”. Both forms appear in the main text.

Line 57: The authors affirm “They are host defense peptides…” Please rephrase, not all AMP are HDPs. For instance, synthetic computationally designed peptides and peptidomimetics with improved therapeutic potential are not HDPs.

Line 57 – 59. The authors mention only membrane-associated mechanisms of action. It would be interesting to add a short sentence explaining that some AMPs can also translocate across bacterial membranes to act on intracellular targets, including ribosomes. Non-lytic AMPs may include, buforin II, indolicidin, drosocin, among others.

A native English speaker should revise the manuscript. There are some minor issues.

In some sections of the manuscript, there are repetitive statements. For instance “the overuse and inappropriate prescribing of antibiotics has led to the emergence of more resistant bacteria and toxicity problems”. Although this information is relevant, the authors should carefully check the entire manuscript to modulate the use of repetitive sentences.

This work is focused on AMPs against bacterial infections. It was quite strange for me to see that the terms “germs” and “germs infections” were overused in the manuscript. These terms are too broad in my view and the authors should consider being more specific (e.g., bacterial, fungi, virus, etc.).

Lines 130 – 131: “The less developed body’s immune defence may then increase infection risk in a case of infectious germ transmission”. Very confusing sentence, please rephrase.

Lines 133 to 136 lack literature citations. Please add.

Figure 1: Why this particular database? When I clicked to check the database it appears: server not found. Considering that the authors were very clear on how they used Google Scholar, Pubmed, etc, to find the most relevant articles for their review, I also encourage them to better organize the data in this figure. Initially, the graph must be edited. For instance, the y axis says “percentage”. But percentage of what? AMP sequences? What is “percentage of diversity” in this case? It is not clear. Moreover, this analysis should be highly improved by comparing different databases, including APD, YAMPD, CAMP among others. I could not find the citation (Mahlapuu et al., 2019) described in the figure 1 caption in the references list. Please double-check.

Avoid expressions like “good antibacterial activity (line 139)”, “good” is too relative.

In the topic “The Archival and the Diversity of AMPs” the authors focused only on naturally occurring AMPs. It should be in the topic title. Actually, the authors affirm in the end of the Introduction “This paper provides an overview on the synthetic and long-lasting AMPs analogues”, but little is really described and discussed about synthetic peptides... Therefore, at some points, the manuscript lacks focus considering the authors’ goals.

Line 193 – 195: Same as comment 1.

Figure 2 must be completely redesigned. The quality is poor. There is not description of the PDB IDs used in the figure. Why did the authors bind to an ion? Why did the authors not specify disulfide bonds? Moreover, this figure can be highly improved if the authors add more AMP structural scaffolds (e.g., cyclotides and other head-to-tail or side-to-side chain cyclic peptides, stapled peptides, among others.)

Line 201: What did the authors mean by rolling indolicidin? I understand that some chemical modifications may favor helical structure in indolicidin-like peptides. However, this is a well-known AMP that cause non-lytic effects on bacterial and presents random coil arrangements… There are many other well-known AMPs that adopt α-helix that would be better examples.

Line 202. The same as above… Defensin is a mix of helix, loops and beta-sheet. Not the best example for α-helical AMPs. The authors should rethink their examples in this entire section.

Line 204. Some studies have reported short β-sheet forming AMPs that do not present disulfide bonds (please check: 10.1002/adfm.201202850, 10.1021/acsinfecdis.9b00073, 10.1016/j.biomaterials.2013.10.053).

Line 206. Same of line 2. It would be better if the authors opened a subsection for mixed structures (e.g, defensins).

Line 216: “Besides, many peptides form their active structure only after” please, replace “form” by “adopt”. Moreover, the authors use as example, indolicidin. Indeed, this AMP changes its structural profile to a “boat-like” structural orientation to translocate bacterial membrane and act on DNA. However, it is not clear in the present manuscript that such structural transition occurs. For someone that it not an expert in AMPs the best examples, by far, would be random coil AMPs in solution that, when in contact with bacterial surfaces, adopt α-helical structures to perform their membrane-associated or intracellular mechanisms of action.

In general the “AMP structure” section should be highly improved. What about cyclic peptides? Different structural scaffolds? And, once again, considering the last sentence of the introduction, I believe the authors should provide the readers with more examples for optimized AMPs, not only naturally occurring ones.

Figure 3 looks much better. A more detailed caption should be organized for this one, describing (very briefly) the mechanisms.

“The Bacterial Resistance Mechanisms towards Antibiotics and the Way AMPs can help” How is this topic different from hundreds of other review articles regarding AMPs? Moreover, some of the statements are controversial. There are studies indicating that bacterial can resist to AMP treatment at sub-lethal doses. Most changes occur in the bacteria surface, efflux pumps production, among others (10.1038/s41598-017-04181-y). Finally, bacterial biofilms have attracted great attention in the last five years. Why did the authors decide not to mention this resistance mechanism, which is currently responsible for >70% of all bacterial infections in humans?

“Advantages of AMPs”: the authors mention AMP synergism with antibiotics as an advantage. Please highlight that such strategy is highly controversial. We are still to fully understand the determinants for AMP/antibiotics synergism and, because of that, some results are positive, some are not, some are effective in vitro, but not in vivo and vice-versa. Therefore, maybe the authors should not emphasize this as an advantage.

“Interestingly, AMPs have no/less side effects for the user, as their use causes zero to only very minimal toxicity to the body when compared with other antibiotics”. This sentence is very confusing, please rephrase. Moreover, the authors give only on example of a peptide “citrus-amp1” with non-toxic effects. Please, add more examples.

“AMPs, which have a low synthetic cost, are small molecules that are” please review this statement carefully. Low synthesis cost compare to what? Chemically modified AMPs (D-amino acids, for instance) can be quite expensive. Moreover, they are more expensive to produce then most antimicrobials already available in the market and, this characteristic has been highlighted as a bottleneck in AMP-based therapies. Finally, small molecules are usually describe with up to 200 Da, which is not the case here (AMPs).

The explanation about the anti-cancer activities of AMPs is not complete…. The authors should also mention that the phospholipids asymmetry in cancer cells provides them with more negatively charged residues in their upper leaflet, thus favoring electrostatic attraction of cationic AMPs.

Lines 365 – 367: “Moreover, the cost of production of AMPs is very high when compared with the production costs of conventional antibiotics” The authors should carefully review the manuscript, as they are often controversial.

The relevance of Table 1 is questionable. Antibiotics and AMPs are highly different classes of antimicrobials. If we check carefully, most studies not even consider their correlation in number of mols for biological experiments. I do believe that AMPs represent a promising strategy, but it is not like they will completely replace antibiotics in a short-term. A more interesting comparison here, would be between naturally occurring and optimized AMPs (which should be the focus of this manuscript according with its title and last sentence of the Introduction).

Additional comments

The manuscript by Baharin et al. offers an overview of the current multi-drug resistant bacterial infection crisis and how antimicrobial peptide could be used as a promising therapeutic strategy. The manuscript is divided into subsections that help the reader to follow the authors’ ideas. Nevertheless, I have minor and major concerns about the manuscript. It is not evident how this review article differs from hundreds of other review articles regarding AMPs in the context of resistant bacterial infections. Moreover, in the end of the Introduction, the authors claim that the paper provides an overview on the synthetic and long-lasting AMPs analogues. However, this was not accomplished in any part of the manuscript, as the authors described (with lack of details) many naturally occurring AMPs, instead of focusing on optimized AMPs and their improved characteristics over their parent peptides. Some figures should be reorganized. Please find below a list of comments that should be addressed:

Line 49 and 51: The authors start with “Today, the number of deaths…” However, they cite a reference from 2018. The authors should consider updating this for a more recent reference.

Please use either “anti-microbial peptides” or “antimicrobial peptides”. Both forms appear in the main text.

Line 57: The authors affirm “They are host defense peptides…” Please rephrase, not all AMP are HDPs. For instance, synthetic computationally designed peptides and peptidomimetics with improved therapeutic potential are not HDPs.

Line 57 – 59. The authors mention only membrane-associated mechanisms of action. It would be interesting to add a short sentence explaining that some AMPs can also translocate across bacterial membranes to act on intracellular targets, including ribosomes. Non-lytic AMPs may include, buforin II, indolicidin, drosocin, among others.

A native English speaker should revise the manuscript. There are some minor issues.

In some sections of the manuscript, there are repetitive statements. For instance “the overuse and inappropriate prescribing of antibiotics has led to the emergence of more resistant bacteria and toxicity problems”. Although this information is relevant, the authors should carefully check the entire manuscript to modulate the use of repetitive sentences.

This work is focused on AMPs against bacterial infections. It was quite strange for me to see that the terms “germs” and “germs infections” were overused in the manuscript. These terms are too broad in my view and the authors should consider being more specific (e.g., bacterial, fungi, virus, etc.).

Lines 130 – 131: “The less developed body’s immune defence may then increase infection risk in a case of infectious germ transmission”. Very confusing sentence, please rephrase.

Lines 133 to 136 lack literature citations. Please add.

Figure 1: Why this particular database? When I clicked to check the database it appears: server not found. Considering that the authors were very clear on how they used Google Scholar, Pubmed, etc, to find the most relevant articles for their review, I also encourage them to better organize the data in this figure. Initially, the graph must be edited. For instance, the y axis says “percentage”. But percentage of what? AMP sequences? What is “percentage of diversity” in this case? It is not clear. Moreover, this analysis should be highly improved by comparing different databases, including APD, YAMPD, CAMP among others. I could not find the citation (Mahlapuu et al., 2019) described in the figure 1 caption in the references list. Please double-check.

Avoid expressions like “good antibacterial activity (line 139)”, “good” is too relative.

In the topic “The Archival and the Diversity of AMPs” the authors focused only on naturally occurring AMPs. It should be in the topic title. Actually, the authors affirm in the end of the Introduction “This paper provides an overview on the synthetic and long-lasting AMPs analogues”, but little is really described and discussed about synthetic peptides... Therefore, at some points, the manuscript lacks focus considering the authors’ goals.

Line 193 – 195: Same as comment 1.

Figure 2 must be completely redesigned. The quality is poor. There is not description of the PDB IDs used in the figure. Why did the authors bind to an ion? Why did the authors not specify disulfide bonds? Moreover, this figure can be highly improved if the authors add more AMP structural scaffolds (e.g., cyclotides and other head-to-tail or side-to-side chain cyclic peptides, stapled peptides, among others.)

Line 201: What did the authors mean by rolling indolicidin? I understand that some chemical modifications may favor helical structure in indolicidin-like peptides. However, this is a well-known AMP that cause non-lytic effects on bacterial and presents random coil arrangements… There are many other well-known AMPs that adopt α-helix that would be better examples.

Line 202. The same as above… Defensin is a mix of helix, loops and beta-sheet. Not the best example for α-helical AMPs. The authors should rethink their examples in this entire section.

Line 204. Some studies have reported short β-sheet forming AMPs that do not present disulfide bonds (please check: 10.1002/adfm.201202850, 10.1021/acsinfecdis.9b00073, 10.1016/j.biomaterials.2013.10.053).

Line 206. Same of line 2. It would be better if the authors opened a subsection for mixed structures (e.g, defensins).

Line 216: “Besides, many peptides form their active structure only after” please, replace “form” by “adopt”. Moreover, the authors use as example, indolicidin. Indeed, this AMP changes its structural profile to a “boat-like” structural orientation to translocate bacterial membrane and act on DNA. However, it is not clear in the present manuscript that such structural transition occurs. For someone that it not an expert in AMPs the best examples, by far, would be random coil AMPs in solution that, when in contact with bacterial surfaces, adopt α-helical structures to perform their membrane-associated or intracellular mechanisms of action.

In general the “AMP structure” section should be highly improved. What about cyclic peptides? Different structural scaffolds? And, once again, considering the last sentence of the introduction, I believe the authors should provide the readers with more examples for optimized AMPs, not only naturally occurring ones.

Figure 3 looks much better. A more detailed caption should be organized for this one, describing (very briefly) the mechanisms.

“The Bacterial Resistance Mechanisms towards Antibiotics and the Way AMPs can help” How is this topic different from hundreds of other review articles regarding AMPs? Moreover, some of the statements are controversial. There are studies indicating that bacterial can resist to AMP treatment at sub-lethal doses. Most changes occur in the bacteria surface, efflux pumps production, among others (10.1038/s41598-017-04181-y). Finally, bacterial biofilms have attracted great attention in the last five years. Why did the authors decide not to mention this resistance mechanism, which is currently responsible for >70% of all bacterial infections in humans?

“Advantages of AMPs”: the authors mention AMP synergism with antibiotics as an advantage. Please highlight that such strategy is highly controversial. We are still to fully understand the determinants for AMP/antibiotics synergism and, because of that, some results are positive, some are not, some are effective in vitro, but not in vivo and vice-versa. Therefore, maybe the authors should not emphasize this as an advantage.

“Interestingly, AMPs have no/less side effects for the user, as their use causes zero to only very minimal toxicity to the body when compared with other antibiotics”. This sentence is very confusing, please rephrase. Moreover, the authors give only on example of a peptide “citrus-amp1” with non-toxic effects. Please, add more examples.

“AMPs, which have a low synthetic cost, are small molecules that are” please review this statement carefully. Low synthesis cost compare to what? Chemically modified AMPs (D-amino acids, for instance) can be quite expensive. Moreover, they are more expensive to produce then most antimicrobials already available in the market and, this characteristic has been highlighted as a bottleneck in AMP-based therapies. Finally, small molecules are usually describe with up to 200 Da, which is not the case here (AMPs).

The explanation about the anti-cancer activities of AMPs is not complete…. The authors should also mention that the phospholipids asymmetry in cancer cells provides them with more negatively charged residues in their upper leaflet, thus favoring electrostatic attraction of cationic AMPs.

Lines 365 – 367: “Moreover, the cost of production of AMPs is very high when compared with the production costs of conventional antibiotics” The authors should carefully review the manuscript, as they are often controversial.

The relevance of Table 1 is questionable. Antibiotics and AMPs are highly different classes of antimicrobials. If we check carefully, most studies not even consider their correlation in number of mols for biological experiments. I do believe that AMPs represent a promising strategy, but it is not like they will completely replace antibiotics in a short-term. A more interesting comparison here, would be between naturally occurring and optimized AMPs (which should be the focus of this manuscript according with its title and last sentence of the Introduction).

---

## Round 0.2 · Minor Revisions

Dear Dr. Baharin and colleagues:

Thanks for revising your manuscript. The reviewers are very satisfied with your revision (as am I). Great! However, there are a few minor edits to make. Please address these ASAP so we may move towards acceptance of your work.

Best,

-joe

·

Basic reporting

The authors have addressed all my comments in a satisfactory way.

Experimental design

The authors have addressed all my comments in a satisfactory way.

Validity of the findings

The authors have addressed all my comments in a satisfactory way.

Additional comments

The authors have addressed all my comments in a satisfactory way.

Reviewer 2 ·

Basic reporting

Lines 69 and 70: “This review is intended for all scientists and academicians in related fields to foresee
70 AMPs as a therapeutic agent and as a reference for their future related studies.” this sentence is quite confusing. Please rephrase


The topic “The evolution of synthetic AMPs” is not straightforward and there is no “evolution” “chronology” “time line” considering the different methods applied for naturally occurring AMP optimization, thus generating synthetic variants. If the authors retrieved published that from the last 10 years, in this topic they should start with the rational design strategies applied in 2011 and how they have “evolved” up to date.

The topic “the analogy between naturally occurring AMPs and synthetic AMPs” is really necessary? There is no useful information there, apart from the fact that Table 1 is cited. I suggest just citing Table 1 in another section and removing this topic.

The manuscript still lacks a better connection between all the ideas presented. The writing was improved, but the topics are, somewhat, disconnected. For instance, one can read the “AMP structures” topic without reading the full manuscript, as the topic is loosely connected with the others. This is also very clear in the Conclusions topic, which does not address or summarize all the topics present in the manuscript. I believe the manuscript would benefit of such connections of ideas.

The figures were highly improved

There are still some English issues that must be checked.

Experimental design

Lines 69 and 70: “This review is intended for all scientists and academicians in related fields to foresee
70 AMPs as a therapeutic agent and as a reference for their future related studies.” this sentence is quite confusing. Please rephrase


The topic “The evolution of synthetic AMPs” is not straightforward and there is no “evolution” “chronology” “time line” considering the different methods applied for naturally occurring AMP optimization, thus generating synthetic variants. If the authors retrieved published that from the last 10 years, in this topic they should start with the rational design strategies applied in 2011 and how they have “evolved” up to date.

The topic “the analogy between naturally occurring AMPs and synthetic AMPs” is really necessary? There is no useful information there, apart from the fact that Table 1 is cited. I suggest just citing Table 1 in another section and removing this topic.

The manuscript still lacks a better connection between all the ideas presented. The writing was improved, but the topics are, somewhat, disconnected. For instance, one can read the “AMP structures” topic without reading the full manuscript, as the topic is loosely connected with the others. This is also very clear in the Conclusions topic, which does not address or summarize all the topics present in the manuscript. I believe the manuscript would benefit of such connections of ideas.

The figures were highly improved

There are still some English issues that must be checked.

Validity of the findings

Lines 69 and 70: “This review is intended for all scientists and academicians in related fields to foresee
70 AMPs as a therapeutic agent and as a reference for their future related studies.” this sentence is quite confusing. Please rephrase


The topic “The evolution of synthetic AMPs” is not straightforward and there is no “evolution” “chronology” “time line” considering the different methods applied for naturally occurring AMP optimization, thus generating synthetic variants. If the authors retrieved published that from the last 10 years, in this topic they should start with the rational design strategies applied in 2011 and how they have “evolved” up to date.

The topic “the analogy between naturally occurring AMPs and synthetic AMPs” is really necessary? There is no useful information there, apart from the fact that Table 1 is cited. I suggest just citing Table 1 in another section and removing this topic.

The manuscript still lacks a better connection between all the ideas presented. The writing was improved, but the topics are, somewhat, disconnected. For instance, one can read the “AMP structures” topic without reading the full manuscript, as the topic is loosely connected with the others. This is also very clear in the Conclusions topic, which does not address or summarize all the topics present in the manuscript. I believe the manuscript would benefit of such connections of ideas.

The figures were highly improved

There are still some English issues that must be checked.

Additional comments

Lines 69 and 70: “This review is intended for all scientists and academicians in related fields to foresee
70 AMPs as a therapeutic agent and as a reference for their future related studies.” this sentence is quite confusing. Please rephrase


The topic “The evolution of synthetic AMPs” is not straightforward and there is no “evolution” “chronology” “time line” considering the different methods applied for naturally occurring AMP optimization, thus generating synthetic variants. If the authors retrieved published that from the last 10 years, in this topic they should start with the rational design strategies applied in 2011 and how they have “evolved” up to date.

The topic “the analogy between naturally occurring AMPs and synthetic AMPs” is really necessary? There is no useful information there, apart from the fact that Table 1 is cited. I suggest just citing Table 1 in another section and removing this topic.

The manuscript still lacks a better connection between all the ideas presented. The writing was improved, but the topics are, somewhat, disconnected. For instance, one can read the “AMP structures” topic without reading the full manuscript, as the topic is loosely connected with the others. This is also very clear in the Conclusions topic, which does not address or summarize all the topics present in the manuscript. I believe the manuscript would benefit of such connections of ideas.

The figures were highly improved

There are still some English issues that must be checked.

---

## Round 0.3 · accepted · Accept

Dear Dr. Baharin and colleagues:

Thanks for revising your manuscript based on the concerns raised by the reviewer. I now believe that your manuscript is suitable for publication. Congratulations! I look forward to seeing this work in print, and I anticipate it being an important resource for groups studying antimicrobial peptides as therapeutics treating for infectious by antibiotic-resistant pathogens. Thanks again for choosing PeerJ to publish such important work.

Best,

-joe